# Experimental, Spectroscopic, and Computational Insights into the Reactivity of “Methanal” with 2-Naphthylamines

**DOI:** 10.3390/molecules28041549

**Published:** 2023-02-06

**Authors:** Martin Havlík, Tereza Navrátilová, Michaela Drozdová, Ameneh Tatar, Priscila A. Lanza, Diego Dusso, Elizabeth Laura Moyano, Carlos A. Chesta, Domingo Mariano A. Vera, Bohumil Dolenský

**Affiliations:** 1Department of Analytical Chemistry, University of Chemistry and Technology Prague, Technická 5, 166 28 Praha, Czech Republic; 2QUIAMM-INBIOTEC, Department of Chemistry, Facultad de Ciencias Exactas y Naturales, Universidad Nacional de Mar del Plata, Mar del Plata B7602AYL, Argentina; 3INFIQC, Department of Organic Chemistry, Facultad de Ciencias Químicas, Universidad Nacional de Córdoba, Córdoba 5000, Argentina; 4Instituto de Investigaciones en Tecnologías Energéticas y Materiales Avanzados (IITEMA), Universidad Nacional de Río Cuarto (UNRC), Consejo Nacional de Investigaciones Científicas y Tecnológicas (CONICET), Campus Universitario, Río Cuarto 5800, Argentina

**Keywords:** methanal, naphthylamine, quinazoline, Tröger’s base, spiro-Tröger’s base, mechanisms, DFT

## Abstract

The reactions of 2-naphthylamine and methyl 6-amino-2-naphthoate with formalin and paraformaldehyde were studied experimentally, spectrally, and by quantum chemical calculations. It was found that neither the corresponding aminals nor imines were formed under the described conditions but could be prepared and spectrally characterized at least in situ under modified conditions. Several of the previously undescribed intermediates and by-products were isolated or at least spectrally characterized. First principle density functional theory (DFT) calculations were performed to shed light on the key aspects of the thermochemistry of decomposition and further condensation of the corresponding aminals and imines. The calculations also revealed that the electrophilicity of methanal was significantly greater than that of ordinary oxo-compounds, except for perfluorinated ones. In summary, methanal was not behaving as the simplest aldehyde but as a very electron-deficient oxo-compound.

## 1. Introduction

Although it is generally believed that the chemical reactions of methanal (**1**) with arylamines are already well studied, the opposite is true. The reactivity of methanal has been studied since the days of alchemy, and many of its reactions were described when the theory of bonds was born with the discovery of molecular structures. In those days, scientific methods for determining molecular structure were limited to elemental analyses (EA), which even nowadays have a rather limited accuracy (approx. ±0.3%). Indeed, these analyses are, therefore, incapable of distinguishing compounds with similar elemental compositions and are useless for recognizing isomers. Although the art of the early pioneers of chemistry was enthralling, contemporary scientific knowledge was insufficient to design the molecular structure of isolated products correctly.

Unfortunately, the inaccuracies that arose during these times have only occasionally been corrected, even at a time when nuclear magnetic resonance (NMR) spectroscopy was already a widely available technique for proving the molecular structure of organic compounds. For example, in our previous study [1] on the reactivity of 2-naphthylamine (**2a**) with a methanal equivalent under acidic conditions (Figure 1), we isolated not only the known Tröger’s base (TB) [2,3,4,5,6] derivative **3a** and its methylated side products, but we also discovered its constitutional isomer **4a** named spiro-Tröger’s base (spiroTB). Through a literature survey, we found that TB **3a** was prepared by Reed in 1886 [7,8,9], i.e., a year before Tröger published his base; both studies lacked a correct estimation of the molecular structure of the base. Indeed, SpiroTB **4a** was not discovered by Farrar in 1964 [10] or Margitfalvi in 1998 [11]. In addition to the above-mentioned TB isomers, other described reaction products have included acridine **5a**, quinazoline **6a**, and dihydroquinazoline **7a**, based on the reaction conditions used.

Further ambiguities arise from the fact that the molecular structures of the reaction intermediates have not been accurately identified. Thus, if the true intermediate has a reactivity that is analogous to the expected one, the expected molecular structure of the intermediate may be mistaken for the correct one. This is the case of methanal itself.

Methanal is a very reactive gas with a boiling point of 19.3 °C. It is usually considered the simplest organic aldehyde, but its reactivity differs significantly from ordinary aldehydes. Methanal is more similar to electron-deficient aldehydes such as trichloro- or trifluoroacetaldehyde, which form rather stable hydrates, hemiacetals, and hemiaminals in the presence of water, alcohols, and amines, respectively. The reactivity of methanal is even more unique since it contains no bulky groups attached to a carbonyl group; its low steric hindrance is probably the main cause of oligomerization, and thus, hydrates or hemiacetal of methanal dimer, trimer, etc., are well known. In addition, methanal can undergo a Cannizzaro reaction, i.e., it can act both as a reducing agent as well as an oxidizing one. The ability of methanal to act as a dehydrogenation agent could lead to the formation of dihydroquinazoline **7a** from quinazoline **6a**; such ability was observed for trifluoropyruvate [12].

Formalin is an aqueous solution of methanal, where methanal spontaneously forms the rather stable hydrate **10** (methanediol; equilibrium constant 2300) [13], followed by higher diols and *O*-methyl diols when methanol is used as a stabilizer. Since the concentration of free methanal is low [14,15] and the reaction products of both methanal and diols can be identical, there is no compelling reason to describe the formalin reaction as the reaction of methanal.

Similarly, paraformaldehyde is a mixture of higher diols with the general formula HO-(CH_2_O)_n_-H. Contrary to sources in the literature, boiling paraformaldehyde in methanol or ethanol does not produce methanal but mainly *O*-methyl or *O*-ethyl diols [16]. Pure methanal can only be generated through the dry decomposition of paraformaldehyde [14,15].

In this article, we focus on the conversion of naphthylamine **2a** and methyl 6-amino-2-naphthoate (**2b**) into the corresponding aminals **8a** and **8b**, and imines **9a** and **9b**, respectively (Figure 2). Both aminal **8a** and imine **9a** have been described as the products of the treatment of naphthylamine **2a** with formalin or paraformaldehyde, however, without any spectral evidence of their molecular structures. In light of the current knowledge, their formation under described conditions seems questionable. Since these compounds and their analogs are key intermediates for various syntheses, we have decided to re-examine their preparation.

The calculations presented here form part of our ongoing efforts to identify the best pathways for obtaining Tröger and spiro-Tröger species. A study of the mechanisms leading to the synthesis of these compounds, starting from anilines, has recently revealed important aspects of the mechanisms taking place under strong acid catalysis conditions [17]. The calculations here focus on neutral conditions with acetone as the model solvent.

## 2. Results and Discussion

### 2.1. Studies of Formalin and Paraformaldehyde

First, we studied the ability of formalin and paraformaldehyde to act as a methanal source in a solution using ^1^H, ^13^C, DQF-COSY, HSQC, and HMBC NMR spectra. It is worth noting that the true methanal content was greater than observed because part of it was probably in the gas phase above the solution in the NMR tube. The ^1^H NMR spectra were not recorded under quantitative conditions so that determined contents may have varied within +/−10%.

In the solution of 5 μL of formalin in 0.5 mL of DMSO-*d_6_* (dimethyl sulfoxide), we unambiguously identified (Figure 3) methanal, methanol, water, diols **10**, **11**, and **12**, hemiacetals **13** and **14**, and acetal **15** in a molar ratio of 4:14:783:100:35:10:40:12:3, in addition to traces of formic acid and higher diols and hemiacetals. This corresponded to approximately 12% m/m of methanol in formalin (10–15% was declared by the supplier) and more than 33% m/m of methanal forms (37% was declared by the supplier). In accordance with a similar study in D_2_O [14,15], we did not observe any 1,3,5-trioxane (metaformaldehyde). On the other hand, small amounts of methanal and acetal **15** were observed, which were not mentioned in the study [14,15]. The relatively low concentrations of free methanal (>2%) increased dramatically at higher temperatures, as summarized in Table 1.

The dissolution of 1 mg of paraformaldehyde in 0.5 mL DMSO-*d_6_* with 10 μL of water produced the equilibrated solution, which contained methanal, water, and diols **10**, **11**, and **12** in a molar ratio of 5:9494:441:51:9, alongside traces of higher diols. The methanal contents increased with increasing temperatures while the diols decomposed, as summarized in Table 2. Analogous results were obtained when methanol or ethanol was added to DMSO-*d_6_* instead of water.

Similarly, high relative concentrations of methanal were identified when paraformaldehyde was treated with DMSO-*d_6_* without any added water, so the majority of the paraformaldehyde remained undissolved (both the solvent and the paraformaldehyde contained moisture). The equilibrated mixture in the solution contained methanal, water, and diols **10**, **11**, and **12** in a molar ratio of 61:9038:727:159:15 (the concentrations of higher diols were less than 1‰). As expected, the methanal contents increased (>40%) upon raising the temperature as the diols decomposed; however, a new methanal form appeared and became the major one at 115 °C. This form was characterized by the sole singlet at 4.81 ppm in the ^1^H NMR spectrum (^1^*J*_HC_ = 165.5 Hz, from ^13^C satellites), which was correlated in both the HSQC and the HMBC spectra to a ^13^C singlet at 89.07 ppm. Unfortunately, the molecular structure was not determined; 1,3,5-trioxane has an identical HSQC/HMBC pattern; however, different chemical shifts as was confirmed by the 1,3,5-trioxane standard addition.

It is worth emphasizing that the composition of those samples strongly depended on the water contents, total concentrations, presence of formic acid (common impurity of aged paraformaldehyde), and the time required to reach equilibrium, and thus, reproducing of these experiments would be difficult.

### 2.2. Calculations on the Reactivity of Methanal

As a first task, it was worth quantitatively addressing the power of methanal as a nucleophile, which was also related to its strong tendency to form the hydrate **10**. The reactivity of a species is traditionally related to its HOMO/LUMO (highest energy occupied molecular orbital/lowest energy unoccupied molecular orbital) gap. Thus, methanal was compared with ethanal, acetone, and the perfluorinated aldehyde and ketone parents. As shown in Table 3, methanal had a smaller HOMO/LUMO gap than the heavier parents, and it was close to the gaps found for the perfluorinated oxo-species. 

A more specific and well-established indicator of electrophilicity is available through the density functional theory (DFT) since it could be derived from the total density (i.e., the all-electron wavefunction) instead of the orbital energies (i.e., one-electron wavefunctions). Thus, the electrophilicities were calculated at the most accurate level, using the total energies of the neutral, radical anion, and radical cation [18]: $$\omega =\frac{\mu^{2}}{2\eta }=\frac{{\left(VIP+VEA\right)}^{2}}{2\left(VIP-VEA\right)}$$
where *μ* and η stand for the chemical potential and the chemical hardness: *VIP* and *VEA* are the vertical ionization potential and electron affinities, respectively. Considering that negative electron affinities were involved, the *VEA* and *VIP* were obtained using the methodology proposed by Puiatti et al. [19,20]. According to the results summarized in Table 4, the calculated ω values suggest that methanal was a very strong electrophile.

The higher electrophilicity of methanal also accounted for its ease of hydration, which is well-known in water. Table 5 reports the ΔG°_hyd_ of methanal in a moderately polar solvent, using the ε of acetone as the IEFPCM model solvent [21]. Once again, methanal was found to be more similar to the electron poorer partner than to ethanal.

### 2.3. Attempts to Prepare Aminal ***8a***

Next, we revisited the preparation of aminal **8a**, which was reported in 1902, as the product of the treatment of naphthylamine **2a** with formalin in a molar ratio of 2:1 in acetone under reflux for five hours [22]. The obtained product (a yield was not given) was identified through elemental analyses (84.73 %C, 6.04 %H, and 9.25 %N) as aminal **8a** (calcd. EA: 84.53 %C, 6.08 %H, 9.39 %N) and characterized as having a melting point of 104 °C (from ethanol).

The described procedure was then repeated; however, according to the 1D and 2D NMR spectra, the crude product mainly contained the starting naphthylamine **2a**, quinazoline **6a**, dinaphthylamine **17a**, and bisquinazoline **16a** in a molar ratio of 48:48:3:1, in addition to numerous trace products (Figure 4). The presence of the expected aminal **8a** could be deduced based on the ^1^H NMR signal at 4.70 ppm, which was correlated in the HSQC spectrum with the ^13^C signal at 53.16 ppm, and in the HMBC spectrum with the ^13^C signal at 146.03 ppm. No clear signals of TB **3a**, acridine **5a**, or bisnapthylamine **18a** were observed. The crystallization of the crude product from ethanol allowed us to obtain an insoluble part that contained quinazoline **6a** (2% yield) and bisquinazoline **16a** (3% yield) in addition to crystals of pure quinazoline **6a** (35% yield). The calculated EA for both quinazoline **6a** (85.13 %C, 5.85 %H, 9.03 %N) and bisquinazoline **16a** (85.41 %C, 5.73 %H, 8.85 %N) was not far from the published values. Note that the elemental compositions of dinaphthylamine **17a** and aminal **8a** are identical. Following the subsequent purification and recrystallization of quinazoline **6a** from ethanol, its melting point of 100–102 °C (from ethanol) was measured. Therefore, we conclude that the product published in 1902 [22] consisted of quinazoline **6a** and not aminal **8a**.

The reaction was then repeated under the same condition; however, the crude product contained naphthylamine **2a**, quinazoline **6a**, and dinaphthylamine **17a** in a slightly different molar ratio of 41:45:14, and surprisingly, no NMR signals of either bisquinazoline **16a** or aminal **8a** were identified. Owing to greater concentrations of dinaphthylamine **17a**, we were able to assign all its ^1^H and ^13^C NMR signals and determine its molecular structure. Then, unlike in the original processing [22], we used column chromatography on silica instead of crystallization. Surprisingly, three fractions of various compositions were obtained. The ^1^H NMR spectra were then used to calculate the total yields of the expected starting naphthylamine **2a** (30% recovery) and quinazoline **6a** (51% yield), and surprisingly only 1% yield of dinaphthylamine **17a**. Moreover, acridine **5a** (9% yield) and bisnapthylamine **18a** (8% yield), which were present in all fractions, were also isolated alongside TB **3a** (1% yield), which occurred in a single fraction.

Since neither bisnapthylamine **18a** nor acridine **5a** were present in the crude product, and dinaphthylamine **17a** was isolated in a very low yield, a rearrangement of **17a** into **18a** during the chromatography could be considered, followed by the conversion of **18a** into acridine **5a**, since **18a** was the expected intermediate product of **5a** [1,23]. A similar rearrangement and formation of **18b** when exposed to silica or to air and ambient light for prolonged periods have previously been described [24].

It should be emphasized that the formation of quinazoline **6a** and dinaphthylamine **17a** was surprising since it requires an attack of an R-CH_2_ moiety on the naphthalene core, which generally requires a process of acid catalysis. However, no acid was added to the reaction, and the presence of acid would have led to the formation of TB **3a** and/or spiroTB **4a**. However, these compounds were not observed in the crude product, but after the chromatography procedure, i.e., silica was acidic enough to catalyze the rearrangement of **17a** into **18a** and to TB **3a** formation.

An inspection of the molecular structure of quinazoline **6a** revealed that it might have formed through the cycloaddition of two molecules of imine **9a**, which could have formed during the reaction. However, the proposed mechanism was improbable, as suggested by our experiments on aminal **8b** and imine **9b** (vide infra).

### 2.4. Attempts to Prepare Aminal ***8b***

When the reaction of [22] was performed with methyl 6-amino-2-naphthoate (**2b**) instead of naphthylamine **2a**, we observed a different behavior. While the reaction mixture was a homogenous solution for **2a**, a white solid precipitated shortly after the addition of formalin to the solution of naphthylamine **2b**, a white solid precipitated before slowly dissolving.

The reaction was repeated again, and the white solid intermediate was isolated by filtration (84% yield of crude aminal **8** after 2 h reflux; only 46% after 8 h reflux). The 1D and 2D NMR techniques allowed us to observe that the DMSO-*d_6_* solution of the white solid contained mainly aminal **8b**, which was in equilibrium with hemiaminal **20b** and starting naphthylamine **2b**. When the sample was heated above 85 °C, the decomposition of both aminal **8b** and hemiaminal **20b** was observed, along with the formation of naphthylamine **2b**, imine **9b**, and traces of methanal (Figure 1). Imine **9b** was clearly recognized by the two doublets with ^2^*J*_HH_ = 16.2 Hz in the ^1^H NMR spectra, which are typical for terminal N = CH_2_ groups. However, the low contents (5% *n*/*n*) and low concentrations of imine **9b** prevented us from identifying all of its NMR signals. When the heating was performed with the addition of water, only negligible amounts of imine **9b** were formed, but a significant decomposition of aminal **8b** and formation of methandiol (**10**) occurred (Figure 5). When the sample was cooled back to 25 °C, the imine **9b** was slowly converted back to aminal **8b** and hemiaminal **20b**, over a few hours. No formation of quinazoline **6b** was observed.

### 2.5. Preparation of Imine ***9a*** under Acidic Conditions

The identification of imine **9b** as an unstable compound in the presence of nucleophiles such as water or naphthylamine **2b** (vide supra) prompted us to prepare imine **9a** for comparison via known procedures.

The attempt to prepare imine **9a** was likely first described in 1902 by Möhlau, who produced it as a product of the treatment of naphthylamine **2b** with formalin in a molar ratio of 1:1 in ice acetic acid (a yield was not given) [22]. The molecular structure was suggested based on its EA (found: 85.36 %C, 6.07 %H, 9.17 %N; calcd. in 1902: 85.16 %C, 5.81 %H, 9.03 %N) and characterized by a melting point of 62–64 °C; no attempts were made to purify the compound.

However, when the procedure was reproduced, the obtained white solid contained quinazoline **6a**, TB **3a**, bisquinazoline **16a**, and acridine **5a** in a molar ratio of 71:23:3:3, contaminated with a few unidentified minor products. However, no ^1^H NMR signals of imine **9a** were observed.

In addition, after the aqueous acid filtrate had been left to stand overnight, a few milligrams of a greenish solid precipitated. The 1D and 2D NMR spectra of the product showed that the solid contained acridine **5a**, bisnaphthylamine **18a** (probably as salt **19a**), TB **3a**, dihydrogenquinazoline **7a**, and quinazoline **6a** in a molar ratio of 39:29:23:6:3; due to the high contents of acetic acid, all these compounds were at least partially protonated. The composition of the NMR sample (solution in DMSO-*d*_6_) changed over one day. Bisnaphthylamine **19a** disappeared while the acridine **5a** contents increased, and the 1:1:1 triplet of a ^14^N-ammonium cation appeared (7.12 ppm, 51.1 Hz); this confirmed the pathway for acridine **5a** formation suggested by [9]. On the other hand, at least two unidentified compounds were formed.

The reaction was then repeated; however, immediately after quenching the reaction by adding water, the ammonium solution was added until basic pH. After the extraction and repeating the column chromatography, we isolated quinazoline **6a** (50% yield), TB **3a** (10% yield), acridine **5a** (3% yield), and the starting naphthylamine **2a** (21% yield), and dihydrogenquinazoline **7a** (3% yield), and surprisingly, also oxo-TB **21a** (2% yield) and oxo-quinazoline **22a** in yield of less than 1% (Figure 6).

The formation of the oxo-compounds **21a** and **22a** as side products of a TB derivative preparation has never been reported. The only known oxo-TB analogs were previously prepared via formylation (via *s*BuLi and DMF) followed by aerial oxidation or through the direct oxidation of TB via KMnO_4_ (9 h reflux in CH_2_Cl_2_) [25]. Hence, the formation of the oxo-compounds **21a** and **22a** under our mild conditions over a two-minute reaction time was surprising. The low yields of the oxo-compounds **21a** and **22a** were relatively high compared to the yields of their possible precursors **3a** and **7a**, respectively. In addition, the most polar fraction resulting from the chromatographic separation was a complex mixture; however, the 1D and 2D NMR spectra revealed a set of signals which could be attributed to an *exo*-diastereomer of hydroxy-TB **23a** formed as a possible intermediate product of oxo-TB **21a**.

### 2.6. Preparation of Imine ***9a*** under Basic Conditions

Imine **9a** formation was previously mentioned by Kadutskii in 2002, 2006, and 2012 through the treatment of naphthylamine **2a** with paraformaldehyde (1:1) in ethanol in the presence of a catalytic amount of NaOH [26,27,28]. However, no spectroscopic data confirming the presence of imine **9a** were reported. The formation of imine **9a** in situ was only assumed based on the molecular structure of the isolated products, which could be considered the result of imine **9a** reactivity. 

Thus, the reaction was repeated and followed by NMR spectroscopy. Our analysis showed that the naphthylamine **2a** was slowly consumed to reach an equilibrium with the majority of the imine-ethanol adduct **24a**, followed by traces of the starting naphthylamine **2a** and probably methanediol (**10**) and minor products (Figure 7). Methanediol (**10**) was identified based on the singlet at 4.61 ppm in the ^1^H NMR spectrum, which was correlated in the HSQC spectrum with a ^13^C signal at 88.83 ppm, and exhibited no correlation in the HMBC spectrum (the chemical shifts may have been strongly affected by the presence of ethanol and sodium hydroxide).

When the sample was heated from 25 to 115 °C, no formation of imine **9a** was observed, possibly due to the large excess of ethanol in the solution. This observation was similar to the heating of aminal **8b** after the addition of water (vide supra).

Thus, a part of the reaction mixture was evaporated to dryness, and the residue was dissolved in 500 μL of DMSO-*d_6_*. A combination of 1D and 2D NMR spectra and mass spectrometry (MS) spectra revealed that the mixture contained not only adducts **24a** and **25a** but also its higher analog **26a**. Newly, the mixture also contained aminal **8a** and its higher analog **27a** (Figure 7). These compounds were not unambiguously proved through NMR spectra due to overlaid signals. Neither methanal nor imine **9a** were observed. However, when the sample was heated to 115 °C, the formation of imine **9a** was observed, and the ^1^H NMR spectrum of imine **9a** was obtained by subtracting the spectra obtained at 40 °C before and after heating (Figure 2). The formation of imine **9a** at high temperatures was in accord with the formation of imine **9b** and methanal (**1**) by heating aminal **8b** and methanal equivalents, respectively (vide supra).

It should also be emphasized that we observed a ^1^H-^13^C HMBC correlation of CH_2_ hydrogen atoms occurring exclusively with nitrogen-bearing carbon (well-separated signals around 145 ppm). This implies that there was no C-alkylation of the naphthalene core under these basic conditions, as is typical for reactions in the presence of an acid (the formation of TB derivatives) or in its absence (the formation of quinazoline). This means that the C-alkylation requires, at least not a too basic condition. In addition, since imine **9a** was formed under these conditions in at least trace amounts, and quinazoline **6a** was not observed as a product, the possible formation of quinazoline **6a** by cycloaddition of imine **9a** is unlikely.

### 2.7. Density Functional Theory Calculations

The processes studied here are summarized in Figure 8, where the relative free energies of the main species refer to the energies of aminals **8a/b** and two methanal molecules as ΔG° = 0. Views of the 3D structures of the main stationary points in Figure 8 are available in the Appendix A.

The thermal decomposition of aminals **8a/b** yielding imines **9a/b** and naphthylamines **2a/b** was found to be slightly endergodic. The entrance of the first methanal to position 1 of the naphtylamine to yield **i1a/b** was found to be slightly exergodic, passing through transition states (TS) at 32.0 and 34.2 kcal/mol of the relative free energies for obtaining **i1a** and **i1b**, respectively. Then, the **i1a/b** species could react with the imine **9a/b** to yield the adduct **i2a/b**, a process that was exergodic by about 1 kcal/mol and with a relatively small activation barrier. The intermediate products **i2a/b** could, in principle, close the quinazoline ring through an intramolecular Diels-Alder-like TS for yielding **6a/b**. However, this process seems unlikely due to the high activation barrier of the **TS-i2-6** compound (Figure 8). A more likely pathway would have been the direct recombination of two imines **9a/b** condensing to a tautomer of **6a/b** through **TS-9-6_tau_**, which had a relative free energy of about 20 kcal/mol lower than the former **TS-i2-6**. The intermediate product could easily tautomerize to **6a/b**, which had remarkable stability of −14.1 and −14.7 kcal/mol for **6a** and **6b**, respectively.

Considering the stabilities of **6a/b**, it is interesting to consider the different fates of this quinazoline. The thermochemistries of the formal dehydrogenation of **6a/b** to yield **7a/b** were found to be 2.3 and 4.7 kcal/mol endergodic in the cases of **6a** and **6b**, respectively. While the current calculations were performed considering a neutral acetone medium, the process has previously been described under different conditions, i.e., in an acid media [23]. Under those conditions, the overall thermochemistry of the process mediated by the reduction of imines **9a/b** or reductive cleavage of **8a/b** is shown separately in Figure 9. Both processes had overall spontaneous thermochemistry by 18–19 and 13–15 kcal/mol. Both reactions leading to the formation of imines **7a/b** were more favored for **6b** than for **6a**.

Another expected process for **6a/b** (Figure 8) was the entrance of another methanal molecule. At this stage, a partner of this quinazoline has already been proven as a key intermediate product formed in the last steps of the formation of TBs in the case of anilines as starting amines [17]. The attachment of the methylene unit involves the formation of an unstable **i3_tau_** intermediate that is rapidly tautomerized to **i3**. The activation barrier was found to be at 32.8 kcal/mol for **6a** and was 2 kcal/mol higher for **6b**. The intramolecular electrophilic attack of the alcohol carbon concerted with the water release (**TS-i3-3**) finally led to the formation of TB **3a/b**. Once again, the process was easier for **i3a** than for **i3b.** The overall thermochemistry leading to the TB **3a/b** starting from aminals **8a/b** was found clearly exergodic by 30–31 kcal/mol.

## 3. Materials and Methods

### 3.1. Measurements and Materials

All chemicals and solvents were purchased from commercial suppliers and used without further purification. The NMR spectra were recorded on a 500 MHz instrument. The chemical shifts (*δ*) are indicated in ppm followed by their multiplicity, integral intensity, corresponding coupling constants (*J*) in Hz, and by the signal assignment, which is based on an analysis of ordinary ^1^H-^1^H COSY, ^1^H-^1^H NOESY, ^1^H-^13^C HSQC, and ^1^H-^13^C HMBC correlation spectra. The 2D spectra were recorded with high-resolution conditions utilizing the nonuniform sampling and processing with a linear prediction; hence the reproducibility of the chemical shifts was a few tens of ppb. The “cov.” in the spectra description means that the signal is seriously covered by others, so a full description was impossible. The ^1^H and ^13^C APT chemical shifts are referenced to TMS (using the solvent signals CHCl_3_ 7.26 ppm, CDCl_3_ 77.0 ppm, CHD_2_SOCD_3_ 2.50 ppm, CD_3_SOCD_3_ 39.52 ppm). The MNOVA software was used for the processing, prediction, and simulation of NMR spectra. The interpretation of most spectra required a simultaneous analysis of various 1D and 2D spectra of two or more samples with different compositions. See also Appendix A. The mass spectra were obtained using electrospray ionization (ESI) or atmospheric pressure chemical ionization (APCI) with a linear quadrupole trap (LTQ) Orbitrap spectrometer. Silica (40–63 D, 60 Å) was used to separate the compounds by column chromatography.

### 3.2. Computational Procedure

The electronic structure calculations were all performed in the Gaussian 16. Rev. A03 package [21]. The structures of the reagents, transition states (TS), intermediaries, and products were all optimized at the level of theory CAM-B3LYP/6-311+G(d,p), with the solvent model SCRF-IEFPCM [29] (acetone). The CAM-B3LYP energies were corrected to consider the dispersion effects using the D3 version of the Grimme’s dispersion with the Becke–Johnson damping, i.e., CAM-B3LYP-GD3BJ [30,31] (uncorrected energies available as Appendix A). The stationary points were characterized by their Hessian matrix, which was diagonalized to obtain the harmonic frequencies and then the zero point corrections to energy, enthalpy, and free energy. In relevant cases, an in vacuo intrinsic reaction coordinate (IRC) was determined using a mass-weighted step of 0.02 atomic units and by recalculating the Hessian every ten or 20 steps. The procedure was previously described and tested elsewhere [17,32]. For the relative free energies of Figure 8, the ΔG° = 0 was taken as the energies of aminals **8a/b** plus two methanal molecules to close the mass balance.

The level of theory and general procedure were previously described for the synthesis of symmetric and asymmetric TBs from anilines as starting reagents [17]. The procedure for obtaining the *VEA*s and *VIP*s to compute the electrophilicity using CAM-B3LYP was described in detail in [20].

### 3.3. Studies of Formalin and Paraformaldehyde

(a)An NMR tube was charged with 500 μL of DMSO-*d_6_* (standard quality) and 5 μL of formalin (ASC reagent, formaldehyde solution, 37% m/m in H_2_O, containing 10–15% methanol as a stabilizer) and closed with a gas-tight cup. The solution was monitored by ^1^H NMR spectra at 25 °C. Equilibrium was reached within a few hours. The sample composition was determined by 1D and 2D NMR (Table 1). The sample was heated to 50 °C, left to equilibrate (1–2 h), and analyzed by ^1^H NMR. The same was done at temperatures of 75, 100, and 115 °C. After cooling back to 25 °C, the compositions were slowly returned to equilibrium (two days).(b)An NMR tube was charged with paraformaldehyde (1.0 mg, 33 μmol), DMSO-*d_6_* (0.5 mL), and water (10 μL, 555 μmol) and closed with a gas-tight cup. The mixture was shaken until the paraformaldehyde dissolved and then monitored by ^1^H NMR at 25 °C. Equilibrium was reached within several hours. The sample composition was determined by 1D and 2D NMR (Table 2); the higher diols contents did not exceed 0.5‰ (*n/n*). The sample was heated following the procedure described in (a).(c)An NMR tube was charged with paraformaldehyde (1.0 mg, 33 μmol) and DMSO-*d*_6_ (0.5 mL) and closed with a gas-tight cup. The mixture was shaken, but part of the paraformaldehyde remained undissolved. The sample was heated following the procedure described in (a).

Methanal (**1**): ^1^H NMR (500 MHz, DMSO-*d_6_*, 25 °C): 9.57 (2H, s, ^1^*J*_HC_ = 178.7). ^13^C{^1^H} NMR (126 MHz, DMSO-*d_6_*, 25 °C): 197.58.

Methandiol (**10**): ^1^H NMR (500 MHz, DMSO-*d_6_*, 25 °C): 5.78 (2H, t, 7.4, OH), 4.59 (2H, t, 7.4, CH_2_). ^13^C{^1^H} NMR (126 MHz, DMSO-*d_6_*, 25 °C): 81.95.

Diol (**11**): ^1^H NMR (500 MHz, DMSO-*d_6_*, 25 °C): 6.11 (2H, t, 7.8, OH), 4.68 (4H, d, 7.8, CH_2_). ^13^C{^1^H} NMR (126 MHz, DMSO-*d_6_*, 25 °C): 84.05.

Diol (**12**): ^1^H NMR (500 MHz, DMSO-*d_6_*, 25 °C): 6.33 (2H, t, 7.8, OH), 4.67 (4H, d, 7.8), 4.78 (2H, s). ^13^C{^1^H} NMR (126 MHz, DMSO-*d_6_*, 25 °C): 86.19 (CH_2_), 84.95 (CH_2_OH).

Hemiacetal (**13**): ^1^H NMR (500 MHz, DMSO-*d_6_*, 25 °C): 6.15 (1H, t, 7.7, OH), 4.53 (2H, d, 7.7, CH_2_), 3.22 (3H, s, CH_3_). ^13^C{^1^H} NMR (126 MHz, DMSO-*d_6_*, 25 °C): 89.46 (CH_2_), 53.92 (CH_3_).

Hemiacetal (**14**): ^1^H NMR (500 MHz, DMSO-*d_6_*, 25 °C): 6.35 (1H, t, 8.0, OH), 4.66 (2H, d, 8.0, CH_2_OH), 4.64 (2H, s, CH_2_), 3.25 (3H, s, CH_3_). ^13^C{^1^H} NMR (126 MHz, DMSO-*d_6_*, 25 °C): 91.71 (CH_2_), 84.81 (CH_2_OH), 54.90 (CH_3_).

Acetal (**15**): ^1^H NMR (500 MHz, DMSO-*d_6_*, 25 °C): 3.23 (6H, s, CH_3_), 4.49 (2H, s, CH_2_). ^13^C{^1^H} NMR (126 MHz, DMSO-*d_6_*, 25 °C): 96.79 (CH_2_), 54.53 (CH_3_).

Water: ^1^H NMR (500 MHz, DMSO-*d_6_*, 25 °C): 3.34 (2H, s).

Methanol: ^1^H NMR (500 MHz, DMSO-*d_6_*, 25 °C): 4.10 (1H, q, 5.1), 3.17 (3H, d, 5.1). ^13^C{^1^H} NMR (126 MHz, DMSO-*d_6_*, 25 °C): 48.60. 

Formic acid: ^1^H NMR (500 MHz, DMSO-*d_6_*, 25 °C): 8.14 (1H, s, CH), the signal of OH was not observed, likely due to the signal broadness. ^13^C{^1^H} NMR (126 MHz, DMSO-*d_6_*, 25 °C): 163.09.

The unknown form: ^1^H NMR (500 MHz, DMSO-*d_6_*, 25 °C): 4.10. ^13^C{^1^H} NMR (126 MHz, DMSO-*d_6_*, 25 °C): 48.60.

1,3,5-Trioxane (a standard): ^1^H NMR (500 MHz, DMSO-*d_6_*, 25 °C): 5.12 (2H, s, ^1^*J*_HC_ = 166.3). ^13^C{^1^H} NMR (126 MHz, DMSO-*d_6_*, 25 °C): 92.88.

### 3.4. Reaction of Naphthylamine ***2a*** with Formalin under Neutral Condition

(a)Aqueous formaldehyde (37%, 0.1 mL, 1.23 mmol) was added to a solution of naphthylamine **2a** (352 mg, 2.46 mmol) in acetone (20 mL). The mixture was refluxed for five hours. The reaction mixture was evaporated to dryness in vacuo, and the residue was analyzed using 1D and 2D NMR experiments. The residue was purified by crystallization from ethanol. An insoluble fraction (18 mg) was obtained, which contained quinazoline **6a** (7 mg, 2% yield) and bisquinazoline **16a** (11 mg, 3% yield), as well as crystals of pure quinazoline **6a** (133 mg, 35% yield).(b)Aqueous formaldehyde (37%, 0.1 mL, 1.23 mmol) was added to the solution of naphthylamine **2a** (352 mg, 2.46 mmol) in acetone (20 mL). The mixture was refluxed for five hours and then evaporated to dryness in vacuo. The obtained solid (387 mg) contained mostly naphthylamine **2a**, quinazoline **6a**, and dinaphthylamine **17a** in a molar ratio of 41:45:14, according to NMR. The solid was purified by column chromatography on silica (dichloromethane/methanol from 1:0 to 4:1) to produce four fractions of various compositions. The yields were calculated based on the ^1^H NMR spectra: 105 mg (30% recovered) of naphthylamine **2a**, 196 mg (51% yield) of quinazoline **6a**, 3 mg (1% yield) of dinaphthylamine **17a**, 31 mg (9% yield) of acridine **5a**, 31 mg (8% yield) of bisnaphthylamine **18a**, and 5 mg (1% yield) of TB **3a**.

Quinazoline **6a**: ^1^H NMR (500 MHz, DMSO-*d_6_*, 25 °C): 7.85 (1H, ddt, 8.4, 1.1, 0.8, H9), 7.77 (1H, ddt, 9.1, 0.8, 0.5, H16), 7.73 (1H, dddd, 8.1, 1.3, 0.8, 0.5, H18), 7.71 (1H, ddt, 8.0, 1.4, 0.7, H6), 7.66 (1H, ddt, 8.2, 1.2, 0.8, H21), 7.55 (1H, dd, 8.7, 2.5, H4), 7.55 (1H, dd, 9.1, 2.5, H15), 7.46 (1H, ddd, 8.3, 6.8, 1.4, H8), 7.39 (1H, dd, 2.5, 0.5, H13), 7.36 (1H, ddd, 8.2, 6.8, 1.3, H20), 7.24 (1H, ddd, 8.1, 6.8, 1.2, H19), 7.23 (1H, ddd, 8.0, 6.8, 1.1, H7), 6.90 (1H, d, 8.7, H3), 6.46 (1H, br t, 3.6, NH), 4.89 (2H, br s, H11), 4.83 (2H, d, 3.6, H12). ^13^C{^1^H} NMR (126 MHz, DMSO-*d_6_*, 25 °C): 146.89 (C14), 141.49 (C2), 134.31 (C22), 131.65 (C10), 128.52 (C16), 128.34 (C6), 127.79 (C17), 127.24 (C4), 127.23 (C18), 127.12 (C5), 126.43 (C8), 126.39 (C21), 126.11 (C20), 122.96 (C19), 121.58 (C7), 120.47 (C9), 119.35 (C3), 118.40 (C15), 110.30 (C13), 109.17 (C1), 59.32 (C12), 47.53 (C11). HRMS (APCI^+^, MeOH): for C_22_H_18_N_2_ calcd. [M + H]^+^ 311.1543, found 311.1543. M.p. 100–102 °C (from ethanol).

Bisquinazoline **16a**: ^1^H NMR (500 MHz, DMSO-*d_6_*, 20.5 °C): 8.05 (2H, br d, 8.5, H9), 7.84 (2H, dd, 8.0, 1.4, H6), 7.74 (2H, d, 8.8, H4), 7.67 (2H, br d, 8.1, H18), 7.58 (2H, d, 9.0, H16), 7.57 (2H, ddd, 8.5, 6.8, 1.4, H8), 7.57 (2H, d, 8.8, H3), 7.53 (2H, br d, 8.2, H21), 7.42 (2H, dd, 9.0, 2.5, H15), 7.40 (2H, ddd, 8.1, 6.8, 1.1, H7), 7.39 (2H, cov., H13), 7.31 (2H, ddd, 8.2, 6.8, 13, H20), 7.22 (2H, ddd, 8.1, 6.8, 1.2, H19), 5.13 (2H, br s, H23), 4.91 (4H, br s, H12), 4.90 (4H, br s, H11). ^13^C{^1^H} NMR (126 MHz, DMSO-*d_6_*, 25 °C): 146.42 (C14), 141.61 (C2), 134.28 (C22), 131.11 (C10), 128.46 (C16), 128.32 (C6), 128.24 (C5), 127.77 (C17), 127.64 (C4), 127.17 (C18), 126.75 (C8), 126.41 (C21), 126.10 (C20), 123.42 (C7), 123.00 (C19), 121.48 (C9), 119.25 (C3), 118.62 (C15), 115.55 (C1), 109.76 (C13), 68.71 (C23), 63.85 (C12), 48.02 (C11).

Naphthylamine **2a**: ^1^H NMR (500 MHz, DMSO-*d_6_*, 25 °C): 7.61 (1H, dtd, 8.1, 1.3, 0.7, H6), 7.57 (1H, br d, 8.7, H4), 7.49 (1H, dtd, 8.3, 1.2, 0.7, H9), 7.26 (1H, ddd, 8.3, 6.8, 1.3, H8), 7.08 (1H, ddd, 8.1, 6.8, 1.2, H7), 6.93 (1H, dd, 8.7, 2.3, H3), 6.81 (1H, ddd, 2.3, 0.8, 0.5, H1), 5.34 (2H, br s, ^15^N satellites ^1^*J*_HN_ = 83.5, NH_2_). ^13^C{^1^H} NMR (126 MHz, DMSO-*d_6_*, 25 °C): 146.64 (C2), 135.00 (C10), 128.46 (C4), 127.45 (C6), 126.32 (C5), 125.83 (C8), 125.01 (C9), 120.83 (C7), 118.39 (C3), 105.79 (C1). HRMS (APCI^+^, MeOH): for C_10_H_9_N calcd. [M + H]^+^ 144.0808, found 144.0809.

Dinaphthylamine **17a**: ^1^H NMR (500 MHz, DMSO-*d_6_*, 25 °C): 7.76 (1H, d, cov., H9), 7.69 (1H, dd, 8.1, 1.4, H6), 7.64 (1H, d, cov., H17), 7.62 (1H, d, cov., H4), 7.62 (1H, d, cov., H20), 7.57 (1H, d, cov., H15), 7.35 1H, ddd, cov., H8), 7.32 (1H, ddd, 8.2, 6.8, 1.3, H19), 7.14 (1H, ddd, 8.1, 6.8, 1.1, H7), 7.11 (1H, ddd, 8.0, 6.8, 1.3, H18), 7.08 (1H, dd, 8.8, 2.3, H14), 7.07 (1H, d, cov., H3), 6.99 (1H, d, 2.3, H12), 5.85 (1H, br t, 4.5, NH), 4.50 (2H, d, 4.5, H11). ^13^C{^1^H} NMR (126 MHz, DMSO-*d_6_*, 25 °C): 147.25 (C13), 145.01 (C2), 135.24 (C21), 133.91 (C10), 128.48 (C4), 128.12 (C6), 127.97 (C15), 127.41 (C17), 127.01 (C5), 126.44 (C16), 126.30 (C8), 125.92 (C19), 125.43 (C20), 121.87 (C9), 120.86 (C18), 120.84 (C7), 119.10 (C3), 118.69 (C14), 110.40 (C1), 102.38 (C12), 38.79 (C11). HRMS (APCI^+^, MeOH): for C_21_H_18_N_2_ calcd. [M + H]^+^ 299.1543, found 299.1544.

Acridine **5a**: ^1^H NMR (500 MHz, DMSO-*d_6_*, 25 °C): 10.64 (1H, br s, H11), 9.44 (2H, ddt, 8.2, 1.2, 0.7, H9), 8.21 (2H, ddd, 9.1, 0.8, 0.5, H4), 8.12 (2H, br d, 7.8, H6), 8.06 (2H, dd, 9.1, 0.8, H3), 7.87 (2H, dddd 8.2, 7.1, 1.4, 0.3, H8), 7.79 (2H, ddd 7.8, 7.1, 1.2, H7). ^13^C{^1^H} NMR (126 MHz, DMSO-*d_6_*, 25 °C): 147.80 (C2), 132.16 (C4), 131.03 (C5), 129.86 (C10), 128.76 (C6), 127.84 (C7), 127.65 (C3), 127.63 (C8), 126.25 (C11), 124.27 (C9), 123.71 (C1). HRMS (APCI^+^, MeOH): for C_21_H_13_N calcd. [M + H]^+^ 280.1121, found 280.1120.

Bisnaphthylamine **18a**: ^1^H NMR (500 MHz, DMSO-*d_6_*, 25 °C): 7.95 (2H, ddt, 8.6, 1.2, 0.7, H9), 7.60 (2H, ddt, 8.0, 1.5, 0.5, H6), 7.45 (2H, br d, 8.7, H4), 7.15 (2H, ddd, 8.6, 6.8, 1.5, H8), 7.07 (2H, ddd, 8.0, 6.8, 1.1, H7), 6.99 (2H, d, 8.7, H3), 5.47 (2H, br s, NH_2_), 4.36 (2H, s, H11). ^13^C{^1^H} NMR (126 MHz, DMSO-*d_6_*, 25 °C): 143.81 (C2), 133.69 (C10), 131.79 (C1), 128.27 (C6), 127.46 (C5), 127.07 (C4), 125.62 (C8), 122.48 (C9), 120.62 (C7), 119.18 (C3), 23.33 (C11).

TB **3a**: ^1^H NMR (500 MHz, DMSO-*d_6_*, 25 °C): 7.76 (2H, ddt 8.0, 1.4, 0.6, H6), 7.74 (2H, ddt 8.5, 1.2, 0.8, H9), 7.69 (2H, br d, 8.8, H4), 7.47 (2H, ddd, 8.5, 6.9, 1.4, H8), 7.38 (2H, d, 8.8, H3), 7.37 (2H, ddd, 8.0, 6.9, 1.2, H7), 4.96 (2H, dd, 16.8, 0.8, H11a), 4.72 (2H, br d, 16.8, H11b), 4.44 (2H, br s, H12). ^13^C{^1^H} NMR (126 MHz, DMSO-*d_6_*, 25 °C): 145.41 (C2), 130.93 (C10), 130.19 (C5), 128.26 (C6), 127.24 (C4), 126.44 (C8), 124.67 (C3), 124.57 (C7), 121.44 (C9), 121.23 (C1), 66.07 (C12), 55.16 (C11). HRMS (APCI^+^, MeOH): for C_23_H_18_N_2_ calcd. [M + H]^+^ 323.1543, found 323.1544. M.p. 208–210 °C (from ethanol).

### 3.5. Reaction of Naphthylamine ***2b*** with Formalin under Neutral Conditions

Aqueous formaldehyde (37%, 0.1 mL, 1.23 mmol) was added to a solution of naphthylamine **2b** (496 mg, 2.46 mmol) in acetone (20 mL). The mixture was refluxed for two hours (a white solid precipitated after 20 min). After cooling to room temperature, the solid was filtered off, washed with acetone, methanol, and diethyl ether, and dried to obtain 428 mg (84% yield) of crude aminal **8b**. The filtrate was evaporated to dryness. According to the ^1^H NMR and HRMS spectra, the obtained solid contained naphthylamine **2b**, quinazoline **6b,** and aminal **8b** in a molar ratio of 81:15:4. When eight hours of reflux was, only a 46% yield of crude aminal **8b** was obtained.

The NMR tube was charged with 1 mg of the crude aminal **8b** and 500 μL of dry DMSO-*d*_6_ and shaken until a solution was produced. The 1D and 2D spectra revealed the presence of aminal **8b**, naphthylamine **2b**, and hemiaminal **20b** in a molar ratio of 71:23:6 at 25 °C. The sample was heated to 100 °C for an hour and monitored by ^1^H NMR until it reached equilibrium. The 1D and 2D spectra revealed the presence of aminal **8b**, naphthylamine **2b**, hemiaminal **20b**, and imine **9b** in a molar ratio of 67:25:3:5, alongside traces of methanal.

Another NMR tube was charged with 0.64 mg of crude aminal **8b**, 10 μL water, and 500 μL of dry DMSO-*d*_6_ and shaken to get a clumsy solution. The 1D and 2D spectra revealed the presence of aminal **8b**, naphthylamine **2b**, and hemiaminal **20b** in a molar ratio of 70:25:5 at 25 °C. The sample was heated to 100 °C for an hour and monitored by ^1^H NMR until an equilibrium was reached before being rapidly cooled back to 25 °C. The 1D and 2D spectra evidenced the presence of aminal **8b**, naphthylamine **2b**, hemiaminal **20b**, and methanediol (**10**) in a molar ratio of 37:43:16:4, as well as traces of methanal and imine **9b**.

Aminal **8b**: ^1^H NMR (500 MHz, DMSO-*d_6_*, 25 °C): 8.36 (2H, d, 1.9, H6), 7.81 (2H, d, 8.9, H4), 7.80 (2H, dd, 8.7, 1.8, H8), 7.63 (2H, d, 8.7, H9), 7.20 (2H, t, 6.7, NH), 7.11 (2H, dd, 8.9, 2.3, H3), 7.06 (2H, d, 2.3, H1), 4.75 (2H, t, 6.7, H11), 3.85 (6H, s, H13). ^1^H NMR (500 MHz, DMSO-*d_6_*, 100 °C): 8.35 (2H, br d, H6), 7.81 (2H, dd, 8.7, 1.8, H8), 7.79 (2H, d, 8.9, H4), 7.63 (2H, d, 8.7, H9), 7.16 (2H, dd, 8.9, 2.3, H3), 7.08 (2H, d, 2.3, H1), 6.78 (2H, br t, 5.8, NH), 4.80 (2H, br t, 5.8, H11), 3.88 (6H, s, H13). ^13^C{^1^H} NMR (126 MHz, DMSO-*d_6_*, 100 °C): 166.16 (C12), 147.29 (C2), 137.08 (C10), 129.87 (C6), 129.59 (C4), 125.10 (C5), 125.07 (C9), 124.65 (C8), 121.91 (C7), 118.28 (C3), 103.37 (C1), 52.20 (C11), 50.98 (C13). HRMS (APCI^+^, MeOH): for C_25_H_22_N_2_O_4_ calcd. [M + H]^+^ 415.1652, found 415.1656.

Naphthylamine **2b**: ^1^H NMR (500 MHz, DMSO-*d_6_*, 25 °C): 8.33 (1H, d, 1.8, H6), 7.77 (1H, d, 8.8, H4), 7.74 (1H, dd, 8.7, 1.8, H8), 7.54 (1H, d, 8.7, H9), 6.99 (1H, dd, 8.8, 2.2, H3), 6.83 (1H, d, 2.2, H1), 5.84 (2H, br s, NH_2_), 3.85 (3H, s, H12). ^1^H NMR (500 MHz, DMSO-*d_6_*, 100 °C): 8.32 (1H, br d, H6), 7.76 (1H, dd, 8.7, 1.8, H8), 7.74 (1H, d, cov., H4), 7.54 (1H, d, 8.7, H9), 7.03 (1H, dd, 8.8, 2.3, H3), 6.89 (1H, d, 2.3, H1), 5.51 (2H, br s, NH_2_), 3.87 (3H, s, H12). ^13^C{^1^H} NMR (126 MHz, DMSO-*d_6_*, 100 °C, from 1D and HSQC): 166.21 (C11), 148.71 (C2), 137.16 (C10), 129.97 (C6), 129.77 (C4), 124.74 (C5), 124.55 (C9), 124.50 (C8), 121.45 (C7), 118.66 (C3), 105.13 (C1), 50.94 (C12). HRMS (APCI^+^, MeOH): for C_12_H_11_NO_2_ calcd. [M + H]^+^ 202.0863, found 202.0863.

Hemiaminal **20b**: ^1^H NMR (500 MHz, DMSO-*d_6_*, 40 °C): 8.37 (1H, br d, 1.8, H6), 7.83 (1H, cov., H4), 7.79 (1H, dd, 8.6, 1.8, H8), 7.63 (1H, d, 8.6, H9), 7.10 (1H, dd, 8.8, 2.3, H3), 7.08 (1H, br t, 6.8, OH), 7.02 (1H, cov., H1), 5.37 (1H, t, 6.3, NH), 4.71 (1H, t, 6.5, CH_2_), a signal for OCH_3_ was not unambiguously recognized. ^13^C NMR (126 MHz, DMSO-*d_6_*, 40 °C): only 66.46 (CH_2_) from HSQC and 148.35 (C2) from HMBC were identified. HRMS (APCI^+^, MeOH): for C_13_H_13_NO_3_ calcd. [M + H]^+^ 232.0968, found 232.0967.

Imine **9b**: ^1^H NMR (500 MHz, DMSO-*d_6_*, 40 °C): 8.63 (1H, br s), 8.15 (1H, d, 8.6), 8.03 (1H, d, 8.6), 7.99 (1H, dd, 8.6, 1.7), 7.78 (1H, d, 16.2), 7.55 (1H, d, 16.2), 7.42 (1H, dd, 8.6, 2.1), ca. 7.1 (1H, cov.), 3.92 (3H, s). ^13^C NMR (126 MHz, DMSO-*d_6_*, 40 °C, from HSQC): 130.93, 131.06, 128.84, 125.80, 122.12, 104.55; other signals were not observed due to low concentrations. HRMS (APCI^+^, MeOH): for C_13_H_11_NO_2_ calcd. [M + H]^+^ 214.0863, found 214.0861.

Quinazoline **6b**: ^1^H NMR (500 MHz, DMSO-*d_6_*, 25 °C): characteristic signals: 4.98 (2H, s), 4.94 (2H, d, 3.3). HRMS (APCI^+^, MeOH): for C_26_H_22_N_2_O_4_ calcd. [M + H]^+^ 427.1652, found 427.1656.

### 3.6. Reaction of Naphthylamine ***2a*** with Formalin under Acidic Conditions

(a)Aqueous formaldehyde (37%, 3.8 mL, 50 mmol) in acetic acid (99%, 7 mL) was added dropwise (1 min) to the solution of naphthylamine **2a** (7.2 g, 50 mmol) in acetic acid (99%, 110 mL) at room temperature. The mixture was stirred for the next 2 min and then poured into a 1% brine solution (20 mL) and stirred for 30 min. The solid was filtered off, washed with water, and dried in vacuo to obtain 9.1 g of the crude product exhibiting no signals for imine **9a** in the ^1^H NMR spectrum. The crystallization of the crude product from ethanol produced white crystals consisting of a mixture of quinazoline **6a** and bisquinazoline **16a** in a molar ratio of 94:6. Repeating the crystallization procedure produced pure quinazoline **6a** (4.70 g, 60% yield) and pure bisquinazoline **16a** (0.24 g, 3% yield).(b)Aqueous formaldehyde (37%, 0.38 mL, 5 mmol) in acetic acid (99%, 1 mL) was added dropwise (1 min) to the solution of naphthylamine **2a** (0.72 g, 5 mmol) in acetic acid (99%, 11 mL) at room temperature. The mixture was stirred for the next 2 min. The mixture was then diluted with water (20 mL) and alkalized with aqueous NH_3_, and the product was extracted with dichloromethane. The organic solution was washed sequentially with water and brine, dried over anhydrous sodium sulfate, and evaporated to dryness in vacuo. The residue was purified by column chromatography on silica (dichloromethane/methanol from 1:0 to 8:2) to produce quinazoline **6a** (390 mg, 50%), TB **3a** (81 mg, 10%), acridine **5a** (23 mg, 3%), starting naphthylamine **2a** (150 mg, 21%), oxo-TB **21a** (17 mg, 2%), dihydroquinazoline **7a** (25 mg, 3%), and oxo-quinazoline **22a** (5 mg, under 1%). Those compounds were followed by a polar fraction that contained spiroTB **4a** and a diastereomer of unidentified hydroxy-TB **23a**.

Oxo-TB **21a**: ^1^H NMR (500 MHz, DMSO-*d_6_*, 25 °C): 8.78 (1H, ddt, 8.6, 1.3, 0.7, H9), 8.13 (1H, d, 8.9, H4), 7.91 (1H, br d, 8.2, H6), 7.87 (1H, br d, cov., H18), 7.86 (1H, br d, cov., H21), 7.81 (1H, d, 8.8, H16), 7.64 (1H, d, 8.9, H3), 7.64 (1H, ddd, 8.6, 7.0, 1.4, H8), 7.56 (1H, ddd, 8.5, 6.8, 1.3, H20), 7.52 (1H, d, 8.8, H15), 7.49 (1H, ddd, 8.1, 6.9, 1.2, H7), 7.48 (1H, ddd, 8.1, 6.9, 1.2, H19), 5.28 (1H, d, 17.3, H13 *^exo^*), 5.06 (1H, dd, 12.6, 1.5, H12a), 5.01 (1H, br d, 17.3, H13*^endo^*), 4.98 (1H, d, 12.6, H12b). ^13^C{^1^H} NMR (126 MHz, DMSO-*d_6_*, 25 °C, from HSQC and HMBC): 173.83 (C11), 154.16 (C2), 140.41 (C14), 135.44 (C4), 131.87 (C10), 131.02 (C17), 130.88 (C5), 130.58 (C22), 128.51 (C6), 128.49 (C18), 128.27 (C8), 127.36 (C16), 126.75 (C20), 126.32 (C9), 125.58 (C19), 125.39 (C7), 124.86 (C15), 124.74 (C23), 122.89 (C3), 121.92 (C21), 115.07 (C1), 64.34 (C12), 51.36 (C13). HRMS (APCI^+^, MeOH): for C_23_H_16_N_2_O calcd. [M + H]^+^ 337.1335 found 337.1339. HRMS (ESI^+^): for C_23_H_16_N_2_O calcd. [M + H]^+^ 337.1335, too low intensity (<5%); calcd. for [M + Na]^+^ 359.1155 found 359.1157 (100%); calcd. [2M + Na]^+^ 695.2418, found 695.2417 (5%); calcd. [3M + Na]^+^ 1031.3680, found 1031.3695.

Dihydroquinazoline **7a**: identified according to the characteristic singlet at 5.46 ppm (2H, s) having an HSQC correlation to the ^13^C signal at 45.28 ppm from the CH_2_ group (an HMBC correlation to the ^13^C signal at 146.66 ppm), and the singlet at 8.09 ppm (1H, br s) having an HSQC correlation to the ^13^C signal at 146.66 ppm from the N=CH-N group. HRMS (APCI^+^, MeOH): for C_22_H_16_N_2_ calcd. [M + H]^+^ 309.1386, found 309.1388.

Oxo-quinazoline **22a**: ^1^H NMR (500 MHz, DMSO-*d_6_*, 25 °C): 9.83 (1H, ddt, 8.6, 1.3, 0.7, H9), 8.71 (1H, s, H12), 8.41 (1H, br d, 8.8, H4), 8.20 (1H, br d, 2.2, H13), 8.14 (1H, br d, 8.0, H6), 8.13 (1H, dq, 8.7, 0.7, H16), ~8.08 (1H, m, H18), ~8.05 (1H, m, H21), 7.83 (1H, d, 8.8, H3), 7.78 (1H, ddd, 8.6, 6.9, 1.6, H8), 7.74 (1H, dd, 8.7, 2.2, H15), 7.72 (1H, ddd, 8.0, 6.9, 1.3, H7), 7.68–7.63 (2H, m, H19 and H20). ^13^C{^1^H} NMR (126 MHz, DMSO-*d_6_*, 25 °C): 160.71 (C11), 150.26 (C2), 148.22 (C12), 136.05 (C4), 135.57 (C14), 132.94 (C22), 132.57 (C17), 131.79 (C5), 130.49 (C10), 128.75 (C6 or C8), 128.73 (C8 or C6), 128.70 (C16), 128.12 (C21), 127.76 (C18), 127.16 (C19), 126.91 (C20), 126.83 (C7), 126.43 (C9), 126.18 (C13), 126.15 (C3), 125.80 (C15), 115.12 (C1). HRMS (APCI^+^, MeOH): for C_22_H_14_N_2_O calcd. [M + H]^+^ 323.1179, found 323.1172. HRMS (ESI^+^): for C_22_H_14_N_2_O calcd. [M + H]^+^ 323.1179, found 323.1182 (70%); calcd. for [M + Na]^+^ 345.0998 found 345.1001 (93%); calcd. [2M + Na]^+^ 667.2105, found 667.2107 (100%); calcd. [3M + Na]^+^ 989.3211, found 989.3223 (23%).

SpiroTB **4a**: ^1^H NMR (500 MHz, DMSO-*d_6_*, 25 °C): 7.85 (1H, m, H6), 7.77 (1H, d, 8.8, H4), 7.73 (1H, m, H9), 7.42 (1H, m, cov., H8), 7.41 (1H, m, cov., H7), 7.39 (1H, m, cov., H21), 7.36 (1H, m, cov., H20), 7.35 (1H, m, cov., H18), 7.34 (1H, m, cov., H3), 7.34 (1H, m, cov., H19), 7.02 (1H, d, 9.8, H16), 6.01 (1H, d, 9.8, H15), 5.09 (1H, d, 17.8, H13a), 4.89 (1H, d, 17.8, H13b), 3.85 (1H, br d, 12.6, H12a), 3.67 (1H, dd, 12.6, 1.8, H12b), 3.44 (1H, br d, 17.2, H11a), 2.93 (1H, d, 17.2, H11b). ^13^C{^1^H} NMR (126 MHz, DMSO-*d_6_*, 25 °C): 166.99 (C14), 144.54 (C2), 143.04 (C22), 133.89 (C16), 131.84 (C10), 131.65 (C17), 130.26 (C5), 129.00 (C20), 128.31 (C6), 128.12 (C18), 127.40 (C4), 127.31 (C19), 126.58 (C15), 126.55 (C8), 124.97 (C3), 124.73 (C21), 124.63 (C7), 122.74 (C1), 122.16 (C9), 73.33 (C13), 48.67 (C12), 38.76 (C11), 35.59 (C23). HRMS (APCI+, MeOH): for C_23_H_18_N_2_ calcd. [M + H]^+^ 323.1543, found 323.1547. M.p. 84–86 °C decomp. (from methanol) did not match any of the bases isolated by Farrar [10].

Unidentified hydroxy-TB **23a**: ^1^H NMR (500 MHz, DMSO-*d_6_*, 25 °C): 5.84 (1H, br d, 5.2, H11), 6.90 (d, 5.2, OH), 4.77 (1H, d, 12.4, 1.6, H12a), 4.40 (1H, d, 12.4, H12b), 4.96 (1H, d, 16.9, H13a), 4.70 (1H, br d, 16.9, H13b). ^13^C{^1^H} NMR (126 MHz, DMSO-*d_6_*, 25 °C): 82.83 (C11), 60.08 (C12), 54.25 (C13). HRMS (APCI^+^, MeOH): for C_23_H_18_N_2_O calcd. [M + H]^+^ 339.1492, found 339.1494.

### 3.7. Reaction of Naphthylamine ***2a*** with Formalin under Basic Conditions

Sodium hydroxide (0.02 g, 0.5 mmol) was added to a mixture of paraformaldehyde (0.30 mg, 10.0 mmol) and naphthylamine **2a** (1.43 g, 10.0 mmol) in ethanol (20 mL). The reaction mixture was heated in a water bath (50 °C) until paraformaldehyde completely dissolved after ca. 5 min. The reaction mixture was monitored by NMR by sampling 30 μL of the reaction mixture into 500 μL of DMSO-*d_6_*. After one day, 2 mL of the reaction mixture was evaporated to dryness and analyzed by NMR and MS. The solid mainly consisted of the imine-ethanol adduct **24a**, aminal **8a**, adduct **25a**, and diamine **27a** in a molar ratio of 46:28:18:8, and several minor products.

Imine **9a**: ^1^H NMR (500 MHz, DMSO-*d*_6_, 40 °C): the characteristic ^1^H NMR signals were obtained through the subtraction of 1D spectra (Figure 2). 7.92 (1H, d, 8.7, H4), 7.91 (1H, br d, 8.1, H6 or H9), 7.90 (1H, br d, 7.2, H6 or H9), 7.77 (1H, d, 16.3, H11a), 7.56 (1H, d, 2.1, H1), 7.51 (1H, ddd, cov., H7 or H8), 7.50 (1H, d, 16.3, H11b), 7.47 (1H, ddd, 8.1, 6.9, 1.2, H7 or H8), 7.36 (1H, dd, 8.7, 2.1, H3). HRMS (APCI^+^, MeOH): for C_11_H_9_N calcd. [M + H]^+^ 156.0808, found 156.0807.

Adduct **24a**: ^1^H NMR (500 MHz, DMSO-*d*_6_, 25 °C, presence of excess ethanol and NaOH traces): 7.65 (1H, br d, 8.2, H6), 7.64 (1H, br d, 8.7, H4), 7.59 (1H, br d, 8.2, H9), 7.31 (1H, ddd, 8.2, 6.8, 1.3, H8), 7.14 (1H, ddd, 8.2, 6.8, 1.2, H7), 7.03 (1H, dd, 8.7, 2.3, H3), 6.97 (1H, d, 2.3, H1), 4.67 (1H, s, H11), 3.47 (2H, q, 7.0, CH_2_CH_3_), 1.10 (3H, t, 7.0, CH_2_CH_3_), the NH signal underwent a fast chemical exchange with the OH signal of ethanol. ^13^C{^1^H} NMR (126 MHz, DMSO-*d_6_*, 25 °C, from HSQC and HMBC, presence of ethanol excess and NaOH traces): 145.19 (C2), 135.05 (C10), 128.48 (C4), 127.47 (C6), 127.35 (C5), 126.02 (C8), 125.75 (C9), 121.64 (C7), 117.76 (C3), 104.92 (C1), 73.98 (C11), 61.15 (CH_2_CH_3_), 15.11 (CH_2_CH_3_).

Adduct **25a**: identified by the correlation patterns in HSQC and HMBC corresponding to N-CH_2_-OCH_2_-OCH_2_CH_3_, i.e., ^1^H/^13^C: 4.98/59.81, 5.02/80.42, 3.52/62.05, 1.14/15.14, resp. HRMS (APCI^+^, MeOH): for C_14_H_17_NO_2_ calcd. [M + H]^+^ 232.1332, found 232.1327.

Adduct **26a**: identified by the correlation patterns in HSQC and HMBC corresponding to N-CH_2_-OCH_2_-OCH_2_-OCH_2_CH_3_, i.e., ^1^H/^13^C: 4.96/58.08, 5.27/65.08, 4.92/79.67, 3.42/62.10, 1.11/15.07, resp. HRMS (APCI^+^, MeOH): the peak of a molecular ion was not observed.

Aminal **8a**: identified by the correlation patterns in HSQC and HMBC corresponding to N-CH_2_-N, i.e., ^1^H/^13^C: 4.72/52.57. HRMS (APCI^+^, MeOH): for C_21_H_18_N_2_ calcd. [M + H]^+^ 299.1543, found 299.1546.

Diamine **27a**: identified by the correlation patterns in HSQC and HMBC corresponding to N-CH_2_-O-CH_2_-N, i.e., ^1^H/^13^C: 5.03/58.64. HRMS (APCI^+^, MeOH): the peak of a molecular ion was not observed.

## 4. Conclusions

According to the careful observations and tedious analysis of the 1D and 2D NMR spectra of mixtures at various temperatures, we corrected the molecular structures of several products reported previously [9,22]. Moreover, we suggested an alternative explanation for the formation of products reported in the recent literature [26,27,28], namely aminal **8a** and imine **9a**. We discovered how to prepare or generate these compounds in situ and obtained their spectral characterizations for the first time.

We have shown that free methanal should be generated through the known dry decomposition of paraformaldehyde [14,15] and used in a nucleophile-free environment. Alternatively, methanal or its imine could be obtained by heating to above 85 °C when nucleophiles are not present in great excess. Once the methanal or imine is generated, it will be stable but will slowly react back to hydrates, hemiaminals, or aminals in the presence of nucleophiles.

Since methanal (**1**) concentrations during a reaction of formalin, paraformaldehyde, trioxane, dimethoxymethane, or others could be very low or even negligible in the presence of a high excess of nucleophiles such as water, alcohols, or amines, we recommend calling them methanal equivalents instead of the usual methanal sources.

Finally, we have found that the formation of TB **3a** and spiroTB **4a** occurs rapidly even in acetic acid and that even silica can enable the formation of acridine **5a**, dinaphthylamine **18a**, TB **3a**, and spiroTB **4a**.

According to our experiments and DFT calculations, methanal is not behaving as the simplest aldehyde as is commonly assumed, but instead, its behavior resembled that of a very electron-deficient oxo-compound such as trifluoroethanal.

Computational calculations in the absence of acid catalysis showed plausible pathways for the formation of a stable intermediate product **6**, which could evolve into imine **9** and even into TB **3**. Under the conditions of the in silico study, both the imines and TBs are thermodynamically more stable than the aminals. Under these conditions, TB **3a** was preferentially formed from **6a** with respect to TB **3b**.

## Data Availability

All data used to support the findings of this study are included within the article and Appendix A. A few examples of simple NMR spectra are provided. The other spectra are available upon request as raw data files since their readability requires computer assistance due to their complexity.

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
