# Peer review of "Experimental, Spectroscopic, and Computational Insights into the Reactivity of “Methanal” with 2-Naphthylamines"

_molecules, 2023, doi:10.3390/molecules28041549_

Round 1

Reviewer 1 Report

The presented manuscript describes a mechanistic investigation of the reaction of formaldehyde and naphthylamine which aims at identification of reaction intermediates. The value of such study is questionable to my opinion as well as the obtained results. It unclear, how authors have correlated the signals in proton NMR of a very complex mixture containing multiple similar species labeled as #10-15 in molar ratio 4:14:783:100:35:10:40:12:3 (and none of the 1D or 2D NMR spectra copies were provided in ESI; the main text contains of fragments of some spectra). It is also unclear how the authors observed the signal of formaldehyde (which is a gas) at 115oC in solution of DMSO. Is it really possible to perform quantitative determination of a compound in a mixture with 5:9494:441:51:9 ratio via routine NMR techniques especially considering the 1mg loading of the sample?  The authors report NMR chemical shifts with 3 decimals for both 1H and 13C spectra, which is beyond the standard accuracy. HRMS data is also reported with one extra meaningless decimal. Considering poor novelty and urgency of this research, lack of experimental details (such as copies of NMR and HRMS spectra) and questionable results I cannot recommend this work for publication in Molecules Journal.

Author Response

Ref 1: The presented manuscript describes a mechanistic investigation of the reaction of formaldehyde and naphthylamine which aims at identification of reaction intermediates. The value of such study is questionable to my opinion as well as the obtained results.

Answer 1: The presented results of our research show that the current comprehension of a reactivity of methanal and its equivalents is not entirely correct. Our experimental results in accord with our quantum-chemical calculations provide a much more accurate insight into the methanal chemistry. Since the reactions of methanal or its equivalents are quite frequent (300 articles per year in 2012 to 2022), we believe that the conclusions of our work will have implications both for the design of experiments and for the interpretation of results by tens of scientific groups.

Ref 1: It unclear, how authors have correlated the signals in proton NMR of a very complex mixture containing multiple similar species labeled as #10-15 in molar ratio 4:14:783:100:35:10:40:12:3 (and none of the 1D or 2D NMR spectra copies were provided in ESI; the main text contains of fragments of some spectra).

Answer 2: Yes, it was hard work. We measured many 2D spectra with a high number of scans to reach the necessary detection limits, and with high resolution in both dimensions. These spectra were recorded for many hours (3 to 26 hours each). The simultaneous interpretation of the acquired spectra (DGF-COSY, HSQC, HMBC, NOESY) was also very time consuming (typically 6-30 h) since we often had to compare the spectra of two or more samples with different compositions. We have added a note “The interpretation of most spectra required a simultaneous analysis of various 1D and 2D of two or more samples with different compositions.” in the Section 3.1. Measurements and Materials. Presentation of such spectra in a printed version would be almost unusable due to the low resolution and the need for different scaling for each of them. We recorded more than 750 spectra (5 GB of data), the eventual presentation would grow to many tens of pages. Therefore, we decided and stated in the Data availability statement in the manuscript; "Raw NMR data files are available upon request." On the other hand, although the interpretation of the spectra was complex, it was routine for any expert in structure elucidation by NMR. To clarify this further, we have added some sample spectra to the Supporting Materials “IV) The NMR spectra of pure quinazoline 6a and bisquinazoline 16a, and a few examples of the 2D NMR spectra of the studied mixtures.”, and we have changed the note in the Data availability statement to “All data used to support the findings of this study are included within the article and Supplementary Materials. A few examples of simple NMR spectra are provided. The other spectra are available upon request as raw data files, since their readability requires computer assistance due to their complexity.”

Ref 1: It is also unclear how the authors observed the signal of formaldehyde (which is a gas) at 115oC in solution of DMSO.

Answer 3: Great point. The detection of methanal is possible because the system is at least partially closed. The NMR tube was closed by a gas-tight cap as described in the experimental part. This cap is not absolutely tight, however, the amount of evolved methanal is small (under one milligram); its pressure would be very low. By the other words, it is not a distillation process but a gas-liquid equilibration. To clarify this further, we have added the following note into our manuscript “It is worth noting that the true methanal content was greater than observed because part of it was probably in the gas phase above the solution in the NMR tube.”

Ref 1: Is it really possible to perform quantitative determination of a compound in a mixture with 5:9494:441:51:9 ratio via routine NMR techniques especially considering the 1mg loading of the sample?

Answer 4: That‘s right. It is not possible to record a true quantitative spectrum on our equipment for such different concentrations, and due to gas-solution equilibrium of methanal. On the other hand, the observed decreasing/increasing contents were real, since measured at identical settings. To clarify this further, we have added the following sentence to the previous one (see Answer 3) into the manuscript. “The 1H NMR spectra were not recorded under quantitative conditions, so determined contents may have varied within +/-10%.”

Ref 1: The authors report NMR chemical shifts with 3 decimals for both 1H and 13C spectra, which is beyond the standard accuracy. HRMS data is also reported with one extra meaningless decimal.

Answer 5: Two decimals are sufficient for 1H and 13C spectra and one for 2D spectra of ordinary resolution. However, in our cases of the complex mixtures, three decimals were often necessary to differentiate between contained compounds. On the other hand, the observed chemical shifts depends on the mixture composition (probably due to different pH), thus, in accord with the referee we have removed the third decimal. In the case of 2D spectra, we would prefer to keep two decimals since we recorded the spectra with high digital resolution setting and increased through a linear prediction, thus, we were able to reproduce the chemical shifts with much better reproducibility than 0.1 ppm even on the mixtures. To clarify this further, we added “The 2D spectra were recorded with high-resolution condition utilizing the non‑uniform sampling and processing with a linear prediction; hence the reproducibility of the chemical shifts was a few tens of ppb.” in the Section 3.1. Measurements and Materials. We have removed the extra decimal from the HRMS data.

Ref 1: Considering poor novelty and urgency of this research, lack of experimental details (such as copies of NMR and HRMS spectra) and questionable results I cannot recommend this work for publication in Molecules Journal.

Answer 6: On novelty and urgency, see Answer 1. The NMR and HRMS data have been explained and improved significantly as recommended; see Answers 2, 3, and 4.

Unfortunately, there is mentioned neither specific result nor conclusion, which is questionable. We believe that our results/conclusions are correct since they are based on the experimental evidences, which are consistent with the quantum-chemical calculations, and are consistent with published facts. As the consequence, several unsupported assumptions that have been published both in the long past and in the recent times have been corrected.

On the English improvement

The changes described above were implemented into our manuscript, and then we removed the non-language parts, such chemical shifts and literature, to decrease the price for the English correction (Proof-Reading-Service.com; the certificate is attached). Than we implemented the correction of English into the manuscript – see the file Dolensky-Manuscript-Revision_ON.docx. All changes were accepted in the file Dolensky-Manuscript-Final-2.docx. Also the improved file of Supp. Mat. is attached as the file Dolensky-ESI-Final-2.docx.

Reviewer 2 Report

Experimental, spectroscopic, and computational insights on 2 reactivity of “methanal” with 2-naphthylamines

In section 3.2, the author mentioned they have employed CAM-B3LYP/6-311+G(d,p) and with the 431 solvent model SCRF-IEFPCM to compute energy, enthalpy, and free energy. While it is a well-known hybrid density functional approximation combined with a moderate size basis set (i.e., low computational costs), it is missing London dispersion effects. Because, from a foundational point of view, DFT lacks the ability to provide accurate London dispersion interaction energy. And Non-covalent interactions play a critical role in reaction energetics. So, it would be ideal, from a computational perspective, to compare, (and thus, I suggest) the energetics (which is computed by CAM-B3LYP/6-311+G(d,p) ) with a dispersion corrected functional. That will give a much better picture for establishing the experimental findings. 

Author Response

Ref. 2.: In section 3.2, the author mentioned they have employed CAM-B3LYP/6-311+G(d,p) and with the 431 solvent model SCRF-IEFPCM to compute energy, enthalpy, and free energy. While it is a well-known hybrid density functional approximation combined with a moderate size basis set (i.e., low computational costs), it is missing London dispersion effects. Because, from a foundational point of view, DFT lacks the ability to provide accurate London dispersion interaction energy. And Non-covalent interactions play a critical role in reaction energetics. So, it would be ideal, from a computational perspective, to compare, (and thus, I suggest) the energetics (which is computed by CAM-B3LYP/6-311+G(d,p) ) with a dispersion corrected functional. That will give a much better picture for establishing the experimental findings.

Answer: We wish to thank the reviewer’s suggestion as it could improve the thermochemistry summarized in Schemes 8 and 9. In order to account for the dispersion, the energies obtained with CAM-B3LYP were corrected using the D3 version of the Grimme’s dispersion with the Becke-Johnson damping, i.e., CAM-B3LYP-GD3BJ, in the revised version (see added references [30] and [31]). On the other hand, the electrophilicity of small molecules were not modified since the critical challenge in that case is to account for the right electron affinity. As it was extensively tested in reference [20], several dispersion-corrected functionals have not performed better than CAM-B3LYP with our methodology.

The thermochemistry obtained with the (uncorrected) CAM-B3LYP are deferred as supporting information for comparison purposes. The main changes were some stabilizations (2-6 kcal/mol) of the transition states (indeed, loosely bound species). However, the relative order between them was the same as previously. Thus, the discussion involving Schemes 8 and 9 was subjected to minor changes according to the new thermochemistry including dispersion.

On the English improvement

The changes described above were implemented into our manuscript, and then we removed the non-language parts, such chemical shifts and literature, to decrease the price for the English correction (Proof-Reading-Service.com; the certificate is attached). Than we implemented the correction of English into the manuscript – see the file Dolensky-Manuscript-Revision_ON.docx. All changes were accepted in the file Dolensky-Manuscript-Final-2.docx. Also the improved file of Supp. Mat. is attached as the file Dolensky-ESI-Final-2.docx.

Round 2

Reviewer 1 Report

 The authors have provided sufficent explanation to the issues raised in my review and mofidied the manuscript accordingly.